# Polydatin Inhibits Hepatocellular Carcinoma Cell Proliferation and Sensitizes Doxorubicin and Cisplatin through Targeting Cell Mitotic Machinery

**DOI:** 10.3390/cells12020222

**Published:** 2023-01-04

**Authors:** Umar Farooq, Hao Wang, Jingru Hu, Guangyue Li, Shah Jehan, Jinming Shi, Dangdang Li, Guangchao Sui

**Affiliations:** 1College of Life Sciences, Northeast Forestry University, 26 Hexing Road, Harbin 150040, China; 2Key Laboratory of Cell Proliferation and Differentiation of the Ministry of Education and State Key Laboratory of Membrane Biology, College of Life Sciences, Peking University, Beijing 100871, China

**Keywords:** polydatin, anticancer activity, hepatocellular carcinoma, synergism, differential gene expression, cell mitosis

## Abstract

Polydatin (PD) is a natural compound with anticancer activities, but the underlying mechanisms remain largely unclear. To understand how PD inhibited hepatocellular carcinoma (HCC), we studied PD treatments in HCC HepG2 and SK-HEP1 cells, and normal liver HL-7702 cells. PD selectively blocked the proliferation of HCC cells but showed low toxicity in normal cells, while the effects of doxorubicin (DOX) and cisplatin (DDP) on HCC and normal liver cells were opposite. In the cotreatment studies, PD synergistically improved the inhibitory activities of DOX and DDP in HCC cells but alleviated their toxicity in HL-7702 cells. Furthermore, RNA-seq studies of PD-treated HepG2 cells revealed multiple altered signaling pathways. We identified 1679 Differentially Expressed Genes (DEGs) with over a 2.0-fold change in response to PD treatment. Integrative analyses using the DEGs in PD-treated HepG2 cells and DEGs in a TCGA dataset of HCC patients revealed five PD-repressed DEGs regulating mitotic spindle midzone formation. The expression of these genes showed significantly positive correlation with poor clinical outcomes of HCC patients, suggesting that mitotic machinery was likely a primary target of PD. Our findings improve the understanding of PD’s anticancer mechanisms and provide insights into developing effective clinical approaches in HCC therapies.

## 1. Introduction

Liver cancer is the fifth most common malignancy in humans, and the third most common cause of cancer-related human death globally [1,2]. Hepatocellular carcinoma (HCC) generally arises because of hepatitis and cirrhosis, which accounts for 80–90% of primary liver cancers [3]. Due to its immunosuppressive features, unique metabolic environment and anatomical position, the liver may often be colonized by metastatic tumor cells from adjacent organs, especially the colon [4]. The majority of HCC patients are subjected to palliative care, and curative approaches are used for approximately 30–40% of the cases [5]. Surgical resection and liver transplantation are considered the most effective treatments for liver cancers; however, in most cases, liver cancer patients are diagnosed in their late stages and only 15% can benefit from currently available treatments [6]. Chemotherapy is not considered useful for patients with cirrhosis and major resections [2]. Therefore, it is necessary to develop novel chemotherapeutics to efficiently treat liver cancer patients with reduced toxicity and side effects.

Doxorubicin (DOX) is an anthracycline antibiotic and is used for the treatment of several types of cancers, such as hematological malignancies, liver cancer and several other solid tumors [7]. Due to the onset of dose-dependent and irreversible cardiac toxicity, the clinical utility of DOX is limited [8,9]. It predisposes patients to a high risk for congestive heart failure and increases the mortality rate in cancer survivors from cardiovascular diseases [10]. Similarly, cisplatin (DDP) is an anticancer drug used in the treatment of a broad spectrum of malignancies, such as lymphoma, lung, ovarian and bladder cancers [11,12]. Unfortunately, its clinical use is also restricted as it has severe adverse effects similar to many other therapeutics, and often develops resistance with time [13,14]. It manifests various adverse effects in patients, including neurotoxicity, cardiotoxicity, nephrotoxicity and hepatotoxicity in a dose-dependent manner [15].

Polydatin (PD), also known as piceid, is a natural compound mainly found in the roots of the perennial herb *Polygonum cuspidatum* and grapes [16]. Due to PD’s economic cost and low toxicity to vital organs, research efforts have been made to explore its therapeutic activity and anticancer mechanisms [17]. PD is linked to a wide range of activities in cell and animal models, including anticancer effects and protective actions [18]. It can inhibit cancer cell proliferation and tumor growth and induce apoptosis and cell cycle arrest in a variety of malignant cell types, including breast, lung, liver, ovarian and colorectal cancers [19,20,21,22,23]. PD-based cream also has curative effects to skin rashes of cancer patients under the treatments of EGFR inhibitors and improves their life quality [24]. PD is considered as a mitochondrial protector and promotes fatty acid metabolism in animal models [25,26]. Furthermore, PD has been implicated in the induction of mitochondrial oxidative stress and can counteract 5-flurouracil resistance in colon cancer [27].

Despite the current knowledge about the inhibitory and antiproliferative activity of PD against various cancers, the molecular pathways and essential genes modulated by the anticancer effects of PD remain mostly unclear. In the current study, we determined the anticancer activities of PD against HCC and evaluated the combinatorial treatments of PD with DOX and DDP in HCC cells and normal cells. RNA-seq studies were also conducted to explore the altered signaling pathways and essential genes in PD-treated HepG2 cells, followed by the integrative analyses of the data with those of the differentially expressed genes (DEGs) in a TCGA dataset.

## 2. Materials and Methods

### 2.1. Cell Culture and Reagents

HCC HepG2 and SK-HEP1 cells and human normal liver HL-7702 cells were purchased from the Institute of Biochemistry and Cell Biology (Shanghai, China). SK-HEP1 and HepG2 cells were cultured in MEM medium, and HL-7702 was cultured in RPMI-1640 medium, both of which were supplied with 1% penicillin/streptomycin and 10% fetal bovine serum (FBS) at 37 °C in an atmosphere containing 5% carbon dioxide. PD (95%; Cat# 15721), DOX (≥98%; Cat# D1515) and DDP (≥99%; Cat# P4394) were purchased from Sigma-Aldrich (St. Louis, MO, USA).

### 2.2. Cell Viability Assay

Cell proliferation was examined by WST-1 assay. Briefly, 5 × 10^3^ cells/well were seeded in 96-well plates and incubated overnight. PD, DOX and DDP were individually dissolved in DMSO. In all treatments, DMSO was used as a vehicle control, and the volume of each drug was kept below 0.5% (*v*/*v*) of the culture medium. Cells were seeded in the 96-well plates, cultured overnight, and treated with various concentrations of PD, DOX and DDP. The doses of DOX and DDP were previously described [5,28]. Their concentrations used in the treatments were PD (50, 100, 200, 250, 300 µM) in combination with DDP (2.5, 5, 10, 20, 30 µM) or DOX (0.06, 0.125, 0.25, 0.5, 2.1 µM). The cells were treated for 48 h and then 10 µL of WST-1 reagent (Roche, Indianapolis, IN) was added to each well, followed by incubation at 37 °C for 3 h and then measurement of the absorbance at 450 nm using a microplate reader (Molecular Devices, LLC.). Based on the OD_450 nm_, the viability of cells in each well was evaluated and normalized against the control group. The half maximal inhibitory concentration values (IC_50_) were determined using GraphPad Prism 8.

### 2.3. Transwell Assay

One hundred µL of HepG2 cells with a density of 1 × 10^6^ cells/mL supplied with DMSO, 300 or 600 μM was added on top of the filter membrane in a transwell insert (cat# 3422, Corning), and 600 μL of the complete MEM medium containing 10% of FBS was added into the bottom of the lower chamber in a 24-well plate. After 48 h of culture, the transwell inserts were taken out from the plate and cotton-tipped applicators were used to carefully remove the medium and remaining cells that did not migrated through the top of the membrane. The cells that penetrated the membranes were fixed by 600 μL of 70% methanol for 15 min and then stained by 600 μL of 0.2% crystal violet for 10 min. After being washed and dried, the membranes were examined using an inverted microscope and imaged. ImageJ software was used to evaluate the number of penetrated cells in different fields of view for each treatment.

### 2.4. Evaluation of the Phenotypic Effect of Cotreatment

Combinatorial treatments were designed to determine the inhibition of PD in combination with DOX or DDP in HCC cells. The combination index (CI) for each treatment was determined, and the data were visualized on an isobologram according to previously reported methods [29,30]. For the analysis of PD and DDP cotreatment, cells were cotreated with different concentrations of PD and DDP, followed by IC_50_ calculation. For isobologram analysis, the IC_50_ values were organized so that PD was point A on the Y-axis and DDP on the X-axis. A line was drawn between the two points from the Y-axis to the X-axis. Each point in the isobologram corresponded to an IC_50_ value. The points above, on or beneath the line represent the antagonistic, additive, and synergistic effects, respectively. Data for the PD and DOX cotreatment were generated following the same procedure.

### 2.5. Cell Apoptosis Assay

Apoptosis assays were conducted as previously described [5,28]. First, the cells were seeded in 12-well plates and cultured overnight, followed by the treatments of PD, DOX and DDP alone or in combination, with DMSO as a control. After 48 h of the treatments, the cells were harvested by trypsinization, washed by and resuspended in cold PBS, and stained with 10 µg/mL of propidium iodide (PI) and 10 µg/mL of Annexin V-FITC for 10 min using an Apoptosis Detection Kit (Cat# A211-02, Vazyme Biotech Co. Ltd., Nanjing, China). A flow cytometer (AcuriC6, BD Biosciences, CA) was used to analyze the percentage of apoptotic cells.

### 2.6. RNA Isolation, Transcriptome Analysis and RT-qPCR

HepG2 cells were cultured in 10-cm plates in MEM medium. After 48 h, cells in the logarithmic growth phase were replated in 10 cm plates at a density of 1 × 10^6^ per plate and cultured overnight. The plates were then randomly divided into the control and treatment groups with three dishes in each group: control 1, 2, 3 (C1, C2, C3), and treatment 1, 2, 3 (T1, T2, T3). The cells, at about 80% confluence, were treated with DMSO and 300 μM of PD, respectively, for 24 h, followed by total RNA extraction using the Trizol reagent.

The sequencing libraries were prepared using NEBNext^®^ Ultra™ RNA Library Prep Kit for Illumina^®^ (#E7530L, NEB, Ipswich, MA, USA) following the manufacturer’s recommendations, and index codes were added to attribute the sequences to each sample. The library preparations were sequenced on an Illumina platform, and 150 bp paired-end reads were generated. Sequencing reads were aligned to the reference genome (hg38) using HISAT2 (v2.1.0). The read count for each gene in each sample was determined by HTSeq (v0.6.0), and FPKM (Fragments Per Kilobase Million Mapped Reads) was then calculated to estimate the expression levels of the genes in each sample. The DESeq2 package method was used for screening differentially expressed genes. RNA samples were also analyzed by reverse transcription and quantitative PCR (RT-qPCR) following a standard procedure. Briefly, in a reverse transcription reaction, 1 µg of total RNA and 10 pmol/µL of poly (dT) primer were mixed and incubated at 65 °C for 5 min, followed by the incubation at 4 °C for 2 min. The samples were then immediately transferred to 42 °C and incubated for 30 min, followed by incubation at 4 °C. In the qPCR analyses, LightCycler480 SYBR Green PCR Master Mix was mixed with gene-specific primers and the reaction was carried out using a Roche Lightcycler 480. The data of each gene were normalized against β-actin.

### 2.7. GO/KEGG Enrichment Analysis of the RNA-Seq Data

The differentially expressed genes were screened from the RNA-seq data of HepG2 cells treated with PD. Gene symbols were converted into Entrez ID using the org.Hs.eg.db Bio-conductor R package. GO and KEGG enrichment analyses were performed using the clusterProfiler (v4.2.2) package [31]. The pie chart visualizations and heatmaps of DEGs were generated using ggpubr and pheatmap packages, respectively. The line chart was plotted using ggplot2 package according to the FPKM data of the dataset.

### 2.8. Western Blot Analysis

HepG2 cells treated by PD were washed with ice-cold PBS and lysed in a protein lysate buffer. Protein concentrations were determined using the Bradford protein method. Protein samples (20 μg) were resolved by SDS-PAGE and transferred to nitrocellulose membranes. The membranes were blocked for 1 h at room temperature using 5% nonfat milk in TBST and incubated with a specific primary antibody in TBST overnight at 4 °C. After three washes by TBST, the membranes were incubated with a secondary antibody for 1 h at room temperature, and the immunoreactive bands were visualized using an ECL kit (Tanon, Shanghai, China).

### 2.9. Immunostaining to Visualize Cell Mitosis

HepG2 cells cultured on coverslips were treated by DMSO, 300 or 600 µM of PD for 48 h, washed once by PBS that was prewarmed at 37 °C, and then fixed in 4% paraformaldehyde for 15 min, followed by PBS wash twice. The cells were treated by a permeabilization buffer at room temperature for 15 min in total with gentle shaking, and the buffer was replaced by fresh permeabilization buffer every 5 min. After three five-min washes by PBS with gentle shaking, the cells on the coverslips were incubated with a blocking buffer containing 4% BSA in PBS for 10 min, and then incubated with primary antibodies at 4 °C overnight. After three 15-min washes by PBS and 10 min of blocking, the cells were incubated with a secondary antibody at 37 °C for 1 h. The coverslips finally underwent three 15-min washes by PBS followed by the addition of DAPI. In the inspection of mitotic cells, an AURKB antibody (cat# ab70238, Abcam, Hong Kong, China) was used to detect the positions of kinetochores, an α-tubulin antibody (cat# T6074, Sigma-Aldrich, St. Louis, MO, USA) was used to visualize the microtubules of spindles, and DAPI was used to detect the nuclei. The images were captured by a Leica TCS SP8 STED 3X confocal microscope.

### 2.10. Survival Analysis

The survival data of 369 liver cancer patients of the TCGA-LIHC dataset were obtained from the TCGA database [32]. The clinical data files with the gene expression profiles were processed using DESeq2 correction and merged. The patient data were divided into two groups of high and low gene expression levels, and *p*-value was calculated for these genes to plot the survival curves. The data were visualized using the R survival package based on the *p*-values of gene expression. The survival data of the patients from the TCGA database were used for Cox multiple factor analysis using the survival package of the R software. Afterwards, the Cox function was applied to the construction of the model, the riskScore function was used to mark patients’ survival, and the samples were categorized into high and low risk as the input (RISK). The Survidiff function was used for analyzing the input, which was then processed by the Fit Proportional Hazards Regression Model, and a multiple factor survival curve was drawn. The Receiver Operating Characteristic (ROC) and the Area Under the Curve (AUC) of the RISK were calculated using the survivalROC package of the R software.

### 2.11. Statistical Analysis

All the experiments were performed three times, and the data represent the mean ± standard deviation. R 4.1.2 was used for transcriptome data analysis. Cytoscape software was used for the gene network analysis. A student t-test and one way ANOVA were performed for quantitative analysis using GraphPad Prism 8 (GraphPad, San Diego, CA). A *p*-value less than 0.05 was considered as significant. The criterion for statistical significance was indicated by asterisks (* as *p* < 0.05, ** as *p* < 0.01, *** as *p* < 0.001 and **** as *p* < 0.0001) in the figures.

## 3. Results

### 3.1. PD Exhibited Selective Inhibition against HCC Cells versus Normal Liver Cells

We tested the inhibition of PD, DOX and DDP (Figure 1A) against cancerous (HepG2 and SK-HEP1) and normal (HL-7702) hepatic cells. According to the description of the ATCC (American Type Culture Collection), both HepG2 and SK-HEP1 are cancerous hepatic cell lines that have been used in a wide range of liver cancer studies [33], while HL-7702 is a normal hepatic cell line [34]. All three molecules could reduce the viability of these hepatic cells in a dose-dependent manner (Figure 1B). PD inhibited the proliferation of hepatocellular carcinoma HepG2 and SK-HEP1 cells with IC_50_ values of 361.20 and 393.00 μM, respectively, but displayed much reduced inhibition to the growth of the normal liver HL-7702 cells (IC_50_ = 872.00 μM). On the other hand, DOX and DDP exhibited significantly higher inhibition to HL-7702 cells than that to HepG2 and SK-HEP1 cells (IC_50_ = 0.41 versus 1.32 and 1.22 μM, respectively, for DOX, and IC_50_ = 13.30 versus 44.16 and 25.46 μM, respectively, for DDP) (Figure 1B and Table 1). Based on the relative IC_50_ values, the natural compound PD could selectively kill hepatic cancer cells, but DOX and DDP exhibited strong toxicity against normal liver cells. We also examined whether the PD could affect the migration of the HCC cells. In a transwell assay, PD-treatment could significantly reduce the migration of HepG2 cells (Figure 1C).

### 3.2. PD’s Combination with DOX and DDP Exhibited Synergistic Inhibition to HCC Cells

We analyzed the chemotherapeutic activity of PD in combination with DOX or DDP in HCC cells to explore its potential application in cancer therapies. To determine whether PD could enhance the effectiveness of DOX or DDP against HCC cells, we used subtoxic concentrations of DOX or DDP in combination with PD to treat HepG2, SK-HEP1 and HL-7702 cells (Figure 2 and Table 1). In the combinatorial treatments, we used 50, 100, 250 and 300 µM of PD, together with different concentrations of DOX or DDP. In the presence of PD, both DOX and DDP exhibited significantly decreased IC_50_ values in HepG2 and SK-HEP1 (Table 1), suggestive of their improved inhibitory effects on HCC cells in the presence of PD. For instance, in HepG2 cells cultured in the medium containing 300 µM of PD, compared to the individual treatments, DOX and DDP displayed dramatic decreases of their IC_50_ values (DOX: from 1.32 to 0.12 µM, reduced by 11.0-fold, and DDP: from 44.16 to 2.02 µM, reduced by 21.9-fold); in the SK-HEP1 cells, the presence of 300 µM of PD could also significantly decrease the IC_50_ values of DOX (from 1.22 to 0.39 µM, reduced by 3.1-fold) and DDP (from 25.46 to 4.36 µM, reduced by 5.9-fold) (Table 1). On the other hand, in HL-7702 cells with 300 µM of PD, the IC_50_ fold changes versus individual treatments of DOX (from 0.41 to 0.45 µM, reduced by 0.9-fold) and DDP (from 13.3 to 5.65 µM, reduced by 2.4-fold) were much less than those of the same conditions in the two HCC cell lines. These results strongly indicated that PD could significantly potentiate the inhibitory activity of DOX and DDP in HCC cells.

Using the isobologram and combination index (CI) methods, we analyzed whether there was any synergism in the combination of PD with DOX or DDP. In the isobologram analyses, the IC_50_ values of PD at a constant concentration of 50, 100, 200, 250 or 300 µM combined with different levels of DOX or DDP were obtained.

In HepG2 and SK-HEP1 cells, the IC_50_ data were generally plotted below the lines between the IC_50_ values of the two individual drug treatments on the two axes (Figure 2A,B), indicating that PD combination with either DOX or DDP was synergistic in the two HCC cell lines. However, when tested in HL-7702 cells, most IC_50_ values of combinatorial treatments appeared on or very close to the lines, indicating that the cotreatment of PD with DOX or DDP had additive effects on the normal liver cells (Figure 2C). Using the data of isobologram analyses, we also calculated the combination index (CI) values of the combinatorial treatments (Table 2). Consistent with the results of the isobologram analyses, most of the CI values generated by the combinatorial treatments of PD with DOX or DDP in the two HCC cell lines were much less than 1.0, suggesting synergistic effects of these cotreatments, while the CI values of these combinations in HL-7702 cells were close to or larger than 1.0, implicating additive inhibitory effects (Table 2).

### 3.3. PD Enhanced DOX- and DDP-Induced Apoptosis in HCC Cells

DOX and DDP are generic chemotherapeutics for cancer patients, but their significant side effects have restricted their clinical applications. Because we observed that PD could potentiate the anticancer activity of DOX and DDP, we further determined whether their cotreatments could facilitate the apoptosis of HCC cells and if PD could reduce the adverse effects of the two chemotherapeutics on normal liver cells.

For these purposes, we analyzed apoptotic rates of cells treated by PD (300 µM), DOX (0.2 and 1.5 µM, designated as DOX_L_ and DOX_H_, respectively) and DDP (5 and 30 µM, designated as DDP_L_ and DDP_H_, respectively) either individually or combinatorially. The treatments were designed as PD (300 µM), DOX_L_, DOX_H_, DDP_L_ and DDP_H_ alone, and the combinations of PD with DOX_L_ and DDP_L_, (PD + DOX_L_ and PD + DDP_L_, respectively) with DMSO as a control. After 48 h of the treatments, the cells in each group were collected and stained by Annexin V-FITC and PI reagents, which can stain the cells at early and late stages of apoptosis, respectively. The apoptotic rate of the cells in each group was examined by fluorescence activated cell sorting (FACS) analysis.

In HepG2 and SK-HEP1 cells, the combined treatments of PD with DOX_L_ or DDP_L_ showed significantly higher apoptotic rates than those of the individual DOX_L_ and DDP_L_ treatments (Figure 3B,C). On the other hand, in HL-7702 cells, these cotreatments could markedly reduce the apoptotic rates compared to treatments of DOX_L_ or DDP_L_ alone (Figure 3A). Importantly, in both HepG2 and SK-HEP1 cells, PD sensitized the cells to DOX_L_, and PD + DOX_L_ cotreatments showed apoptotic rates comparable to those of DOX_H_ treatments. Similarly, PD + DDP_L_ cotreatments achieved higher apoptotic rate than that of DDP_H_ treatment in SK-HEP1 cells, while PD could still enhance the apoptotic rate of DDP_L_ in HepG2 cells to an extent lower than that of DDP_H_ (Figure 3B,C). In HL-7702 cells, the two cotreatments only caused about 50% and 70% apoptotic cell death of the corresponding individual treatments (Figure 3A). These data suggested that PD could greatly potentiate the activity of DOX and DDP at their subtoxic dosages in promoting HCC cell apoptosis, but significantly alleviate the toxicity of the drugs in normal liver cells.

### 3.4. Analyses of PD-Induced Differential Gene Expression in HepG2 Cells

To determine the differential expression of genes induced by PD treatment, we used 300 µM of PD or DMSO to treat HepG2 cells for 24 h, extracted the total RNAs, and employed RNA-seq to analyze the transcript profiles. The obtained dataset encompassed the expression of all Poly(A)-containing RNAs, including mRNAs. Therefore, we first analyzed all significantly and differentially expressed genes (designated as DEGs) that showed either up- or down-regulated expression in HepG2 cells treated by PD versus DMSO. After analyzing the data by the DESeq2 method with the 2.0-fold change (FC) as a threshold and a false discovery rate (FDR) of ≤0.05, we discovered a total of 1679 differentially expressed genes with 960 (57.18%) upregulated and 719 (42.82%) downregulated (Figure 4A). The RNA-seq dataset in this study is available in the Gene Expression Omnibus (GEO), with access number GSE207282. In the following analyses, we focused on the studies of these DEGs caused by the PD treatment.

KEGG is a major public database to analyze signaling pathways, and thus can be used to reveal the most important signal transduction and biochemical metabolic pathways involving DEGs. Therefore, we employed KEGG pathway enrichment analysis to determine the changes of key signaling pathways in PD-treated HepG2 cells versus DMSO controls. Based on the numbers and corrected *p*-values of these DEGs, which were adjusted by the Bonferroni correction, the DEGs in PD-treated HepG2 cells were significantly enriched in the hepatocellular carcinoma (HCC) set, TNF signaling pathway, Hippo signaling pathway, steroid biosynthesis pathway, etc. (Figure 4B). The gene names are shown in Appendix A.

We also used comprehensive GO enrichment analysis to determine the functions of the DEGs and their associated biological processes altered in the PD-treated HepG2 cells. The DEGs were put into three independent GO categories, and the functional enrichments were evaluated at corrected *p*-value ≤ 0.05, which was adjusted by the Bonferroni correction. The top 10 most enriched GO terms from each category are presented in Figure 4C, and the gene names are shown in Appendix A. In the Biological Process (BP) category, the DEGs in PD-treated HepG2 cells were significantly enriched in the negative regulation of several processes, including negative regulation of cell growth, cell growth and developmental growth, as well as chemotaxis, cytokine-mediated signaling pathways, etc. In the Cell Component (CC) category, the DEGs were enriched in the GO terms of growth cone, different components related to neuronal development, etc. In the Molecular Function (MF) category, the DEGs were mostly enriched in the GO terms of flavin adenine dinucleotide binding, tubulin binding and the binding activity of several receptors, etc. The heatmap of PD-mediated expression changes of the genes involved in HCC and spindle midzone formation is shown in Figure 4D.

### 3.5. Integrative Analysis between the RNA-Seq Results of PD-Treated HepG2 Cells and a TCGA-LIHC Dataset

Our RNA-seq data indicated that PD treatment could affect multiple signaling pathways. Based on these correlative studies, we asked whether PD altered the expression of any particular gene that plays a key role in hepatic oncogenesis. To answer this question and to evaluate how PD treatment could impact the genes related to clinical outcomes of liver cancer patients, we conducted studies to integrate our RNA-seq dataset from the PD-treated HepG2 cells and a TCGA Liver Hepatocellular Carcinoma (TCGA-LIHC) dataset generated from 370 HCC samples with 50 para-carcinoma tissues [32]. First, we carried out analyses of all DEGs, including both coding and noncoding RNAs, in liver cancers versus para-carcinoma tissues of the TCGA-LIHC dataset and those in PD-treated HepG2 cells versus the DMSO-treated cells to generate two volcano plots (Figure 5A,B). In the volcano plot of the TCGA-LIHC dataset, there are altogether 8577 DEGs with FC > 2 and FDR < 0.05, containing 6835 upregulated and 1742 downregulated genes (Appendix A). In the plot of the RNA-seq data from PD-treated HepG2 cells, there are 1679 DEGs with FC > 2 and FDR < 0.05, including 960 upregulated and 719 downregulated genes (Appendix A).

To evaluate whether PD treatment could alter the expression of essential genes and signaling pathways that play crucial roles in liver cancer development, we carried out integrative analyses for the DEGs in both the PD-treated HepG2 cells and TCGA-LIHC dataset. Our aim was to identify those genes that exhibited reverse gene expression between the two datasets, and some of these DEGs likely contributed to the anticancer activity of PD. As a result, we discovered 305 genes in total. Among them, 242 upregulated and 63 downregulated genes in the PD-treated HepG2 cells showed reduced or increased expression levels, respectively, in liver cancers of the TCGA-LIHC dataset. The altered expression of these DEGs is presented as line charts and heatmaps (Figure 5C and Appendix A). Next, we analyzed these genes by the GO pathway enrichment and KEGG enrichment methods. The most enriched GO terms fell into the pathways related to nucleosome, DNA packaging complex, DNA packaging, spindle midzone, protein-DNA complex, etc. The most enriched KEGG pathways were mostly related to metabolism, such as fatty acid degradation, biosynthesis and metabolism, histidine metabolism and β-alanine metabolism, as well as HCC and transcriptional misregulation in cancers, etc. (Figure 5D). Hence, we selected genes from the above enriched pathways based on the reversely expressed DEGs between both datasets for further investigation.

### 3.6. Effects of PD Treatment on the Spindle Formation of HepG2 Cells

To evaluate potential clinical outcomes and prognostic values of the PD-targeted genes, we analyzed the 305 DEGs obtained from the integrative analyses for the relationship between their expression and clinical outcomes. As a result, we found that the altered expression of 72 DEGs significantly correlated with either improved or reduced overall survivals of HCC patients in the TCGA-LIHC dataset (Appendix A). Among them, five genes that regulate the spindle midzone formation were markedly downregulated in response to PD treatment (Figure 6A), including the mitotic motor protein kinesin family members 14 and 18A (KIF14 and KIF18A), the mitotic motor protein centrosome-associated protein E (CENPE), the mitotic kinase polo-like kinase 1 (PLK1) and the Aurora kinase A (AURKA). The *p*-values of their differential changes in PD-treated cells versus the control fell between 6.28 × 10^−25^ and 1.43 × 10^−21^. The PD-repressed expression of these five genes was confirmed by RT-qPCR (Figure 6B), and PD-mediated downregulation of AURKA and PLK1 was also verified by Western blot analyses (Figure 6C). Because these five genes are involved in spindle midzone formation, we further interrogated whether PD treatment could affect the process of cell mitosis. In immunostaining studies, compared with HepG2 cells treated by DMSO, the cells cultured in medium containing 300 and 600 µM of PD for 24 and 48 h showed increased cases of abnormal spindles with characteristics of spindle multipolarity (Figure 6D,E).

### 3.7. Clinical Prognostic Values of the Spindle-Related Genes Targeted by PD

Among the HCC patients in the TCGA-LIHC dataset, the expression of CENPE, AURKA, KIF14, KIF18A and PLK1 showed significantly negative correlation with the overall survival rates (Figure 7A). Due to the contribution of spindle fiber formation to cell mitosis and proliferation, our integrative analyses of the DEGs in both the RNA-seq dataset from the PD-treated HepG2 cells and TCGA-LIHC dataset derived from cancer patients revealed that the spindle formation is likely the primary target of PD in HCC cells. Additionally, for these five genes playing key roles in spindle formation (Figure 6A and Figure 7A, and Appendix A), we analyzed them using the Cox’s multivariate regression according to the patients’ profiles in the TCGA-LIHC dataset. Using the obtained data, we categorized the patients into low- and high-risk groups (185 and 184 patients, respectively) based on the median risk score, and subsequently plotted Kaplan–Meier survival curves of the two groups of patients. As shown in Figure 7B, the clinical outcomes of the low- and high-risk groups significantly corresponded to the long and short survival rates of the patients, respectively (*p* = 0.00034), indicating that the altered expression of the five spindle midzone-related genes possesses an integrative and multivariate prognostic value. Furthermore, we tested the accuracy of our Cox model using the receiver operating characteristic (ROC) curve, a tool to test the accuracy of Cox models. If the value of the area under the curve (AUC) is greater than 0.5 [35], a Cox model would have an excellent predictive power. The ROC analysis for our data generated an AUC value of 0.669 for 3-year overall survival (Figure 7C), indicating the high accuracy of our Cox model analysis.

Overall, we observed the synergistic inhibition of PD in combination with DOX or DDP and its protective effects on normal cells (Figure 8A). Our RNA-seq studies revealed various altered signaling pathways and DEGs in response to PD treatment (Figure 8B). Furthermore, the integrative analyses of our RNA-seq data and the TCGA-LIHC dataset of liver cancer patients indicated that genes regulating cell mitosis are likely the primary targets of PD (Figure 8B).

## 4. Discussion

Up to 90% of primary liver cancer patients are diagnosed as HCC, with an overall prognostic survival time of six to twenty months [36]. In clinical treatments of HCC, the conventional chemotherapeutics always encounter resistance and become ineffective against cancer cells. Frequently, many chemotherapeutics exhibit higher cytotoxicity to normal cells than that to cancerous cells [37]. Due to late diagnosis of the majority of HCC patients, the treatment options are often quite limited, and usually high doses of chemotherapeutics are needed. Clinical studies demonstrated that the excessive use of chemotherapeutics could cause severe adverse effects to many vital organs, including heart and kidneys [38]. Therefore, it is necessary to develop effective therapeutic strategies with improved inhibitory selectivity to HCC cancer cells and minimal or tolerable side-effects to patients.

Natural compounds have been used for medicinal purposes since ancient times to cure or alleviate various diseases, including cancers. They are used either individually or as adjuvants of conventional drugs to achieve desirable curative effects with significantly lower side-effects than those of many generic therapeutics [39]. PD is a natural compound that can be extracted from various plants, especially Japanese knotweed and *Polygonum cuspidatum*. Accumulating evidence has demonstrated the promising antineoplastic and anti-inflammatory effects of PD using both in vitro and in vivo models [17]. Its anticancer activities have been demonstrated in various types of cancers, such as colon, breast, lung and liver cancers [20,40,41,42]. Consistent with our observation, PD was shown to inhibit the proliferation, invasion and migration of HCC cells, and induce their apoptosis, with the concurrent downregulation of several proliferative genes, including MYC, β-catenin, cyclin D1 and Survivin [42]. A number of studies have revealed the protective effects of PD against various chemotherapeutics and cytotoxic drugs. For example, 2-deoxy-D-glucose (2-DG), a competitor of glucose, inhibits glucose degradation and blocks cancer cell division, but its clinical usage at relatively high doses causes cardiac toxicity [43]. However, the combinatorial use of PD with 2-DG ameliorated this toxic effect on hearts and other vital organs [19]. Meanwhile, PD blocked the PI3K/AKT signaling pathway and synergized the anticancer effects of 2-DG in a xenograft mouse model [19].

Another study demonstrated that PD treatment restricted the glycolytic pathway, substantially improved cardiac markers in an animal study and sensitized pancreatic, colon and breast cancer cells to chemotherapeutics, including DOX, DDP and lapatinib [44]. PD also exhibits antioxidant activity and has been associated with increased lifespan [45]. Although PD is considered a general and strong antioxidant agent, its role in mediating oxidative status depends on cell types. For example, PD increased oxidative stress in nasopharyngeal carcinoma cells and induced their apoptosis, but it decreased oxidative stress in cardiomyocytes and reduced the risk of myocardial infarction [46,47].

Despite the current knowledge regarding the anticancer activity of PD in cancer, its mechanism associated with antineoplastic properties in HCC and its toxicity in normal liver cells are largely unclear. In the current study, we evaluated the anticancer activity of PD using liver cancer HepG2 and SK-HEP1 cells and normal liver HL-7702 cells. We found that PD selectively inhibited the proliferation of HCC cells but showed much reduced cytotoxicity to normal liver cells. On the other hand, our experimental data revealed that the generic chemotherapeutics DOX and DDP exhibited relatively high toxicity in the normal cells compared to that in the HCC cells. Therefore, our data strongly suggested the potential of PD as a cancer therapeutic for clinical applications. However, we also noticed that PD exhibited quite high IC_50_ values (361.20 and 393.00 µM) when inhibiting HepG2 and SK-HEP1 cells, respectively. We had similar observation when evaluating the anticancer activities of ellagic acid and thymoquinone [5,28]. These may restrict PD’s clinical applications in HCC therapies due to the relatively low pharmaceutical effectiveness, despite its praiseworthy selectivity. Therefore, the practical usage of these natural compounds at the current stage would be as ancillary agents to synergistically augment the response of cancer cells to generic chemotherapeutics, such as the DOX and DDP shown in our studies. Additionally, they can also be chemically modified to directionally generate analogs with improved anticancer efficacy.

Previous studies have demonstrated that PD sensitized osteosarcoma cells to doxorubicin and paclitaxel [48,49], increased sunitinib activity against renal adenocarcinoma cells [18] and counteracted 5-Fluorouracil resistance in colon cancer cells [27]. Additionally, PD could also reduce the cardiotoxicity of 2-DG and sunitinib in animal models [18,19]. Therefore, we tested whether PD could sensitize the therapeutic activities of generic chemotherapeutics in HCC cells. In the combinatorial use of PD with subtoxic levels of DOX and DDP, we detected synergistic effects in suppressing HCC cell proliferation at a level similar to the inhibition using high doses of DOX or DDP alone. Importantly, the combination of PD with each chemotherapeutic showed significantly reduced toxicity in normal liver cells. Therefore, our current study revealed a novel and effective strategy of combinatorial treatments using PD and generic chemotherapeutics, which can lead to the inhibition of HCC cell survival with minimal side effects.

Multiple reports have demonstrated PD’s anticancer activity, but a systemic study to provide a panoramic view of PD-regulated signaling pathways and target genes was still needed. Therefore, we carried out RNA-seq studies in PD-treated HepG2 cells and discovered multiple pathways, biological processes or functions potentially targeted by PD, including the spindle midzone formation, an essential step of mitosis.

Mitosis is a key process that is well-controlled in both cell proliferation and differentiation. However, in the process of oncogenesis, due to mutations and altered expression of key regulatory genes, cancer cells typically exhibit deregulated and increased proliferative rates. The integrative analysis of the DEGs in the dataset of PD-treated HepG2 cells and the TCGA-LIHC dataset of HCC patients revealed that five genes regulating spindle fiber formation, AURKA, CENPE, KIF18A, KIF14 and PLK1, were significantly downregulated by PD treatment. We further observed that PD treatment increased the percentage of cells with multipolar spindle formation, an abnormal spindle assembly that could sensitize cancer cells to apoptotic stimuli [50]. Actually, a number of anticancer drugs can provoke spindle abnormality. A previous study demonstrated that ZM447439, a selective Aurora kinase inhibitor, could induce multipolar spindles of laryngeal carcinoma cells and subsequently lead to apoptosis [51]. Paclitaxel, a popular cancer therapeutics, could also promote multipolar mitotic spindle formation that caused high rates of chromosomal instability and elicited cell death [52].

Furthermore, both the univariate and multivariate analyses of these five genes displayed positive correlations between their expression and poor clinical outcomes of HCC patients. Therefore, the spindle formation, a key step of mitosis, is likely a primary biological process targeted by PD.

In response to PD treatment, we observed the perturbations of many additional pathways and biological functions based on the integrative DEG analyses (Figure 5D), especially in several metabolic processes, such as fatty acid biosynthesis and degradation, metabolism of histidine and alanine, etc. It is unclear whether PD caused the alterations of these biological processes directly or indirectly. Future research will be needed to identify the directly associated targets of PD in cells, which will unveil its anticancer mechanism at the molecular level.

## 5. Conclusions

PD inhibits HCC cell proliferation and synergizes DDP and DOX activities against HCC cells, but it protects normal cells by mitigating the toxicity of chemotherapeutics. Our RNA-seq data indicates that PD treatment in HepG2 cells causes the altered expression of genes involved in multiple signaling pathways and biological processes. Due to the importance of mitosis in cell proliferation and the downregulated expression of five genes regulating spindle midzone formation, we propose that the mitotic process is likely a primary target of PD in HCC cells.

## Figures and Tables

**Figure 1 cells-12-00222-f001:**
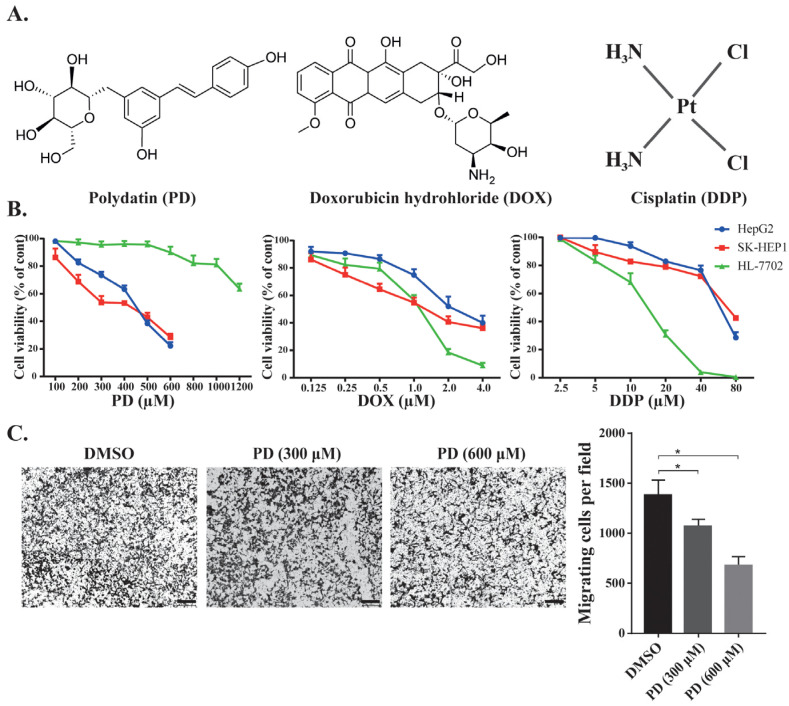
Effects of chemotherapeutic agents in HCC and in normal cell lines. (**A**) Illustration of the chemical structures of PD, DOX and DDP. (**B**) Cell viability of the normal liver HL-7702 cells and HCC HepG2 and SK-HEP1 cells when treated by PD, DOX and DDP. The cells were treated with different concentrations of the chemicals for 48 h, followed by WST-1 assays to evaluate cell proliferation and calculation of cell viability. (**C**) Evaluation of PD’s effects on HepG2 cell migration. HepG2 cells in the transwell were treated by DMSO, 300 and 600 µM of PD for 48 h, followed by staining using crystal violet. Representative images are shown on the left, and the quantitative data from three separate experiments are shown on the right. The scale bar in each image is 100 µm. * as *p* < 0.05.

**Figure 2 cells-12-00222-f002:**
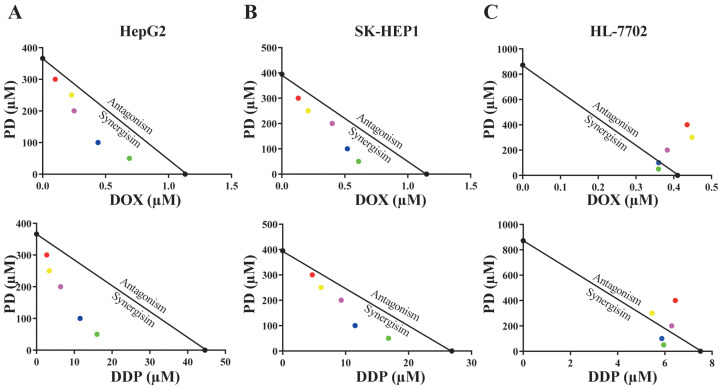
Isobologram analyses of anti-cancer activity of PD and DOX or DDP combinations in HepG2 (**A**), SK-HEP1 (**B**) and HL-7702 (**C**) cell lines using IC_50_ values. The color dots represent various drug concentrations of PD (300 μM-red; 250 µM-yellow; 200 µM-purple; 100 µM-blue; and 50 µM-green).

**Figure 3 cells-12-00222-f003:**
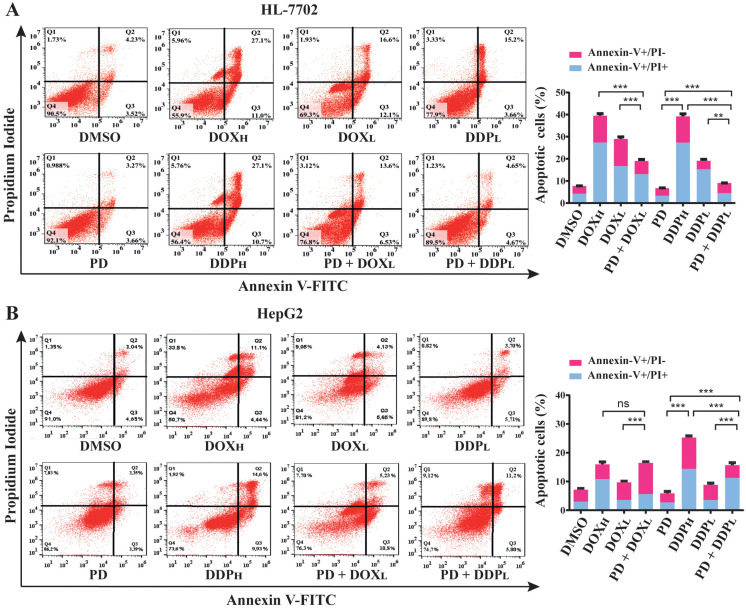
Evaluation of apoptotic rates in normal liver HL-7702 cells (**A**), HCC HepG2 (**B**) and SK-HEP1 (**C**) cells using PD (300 µM) and PD in combination with high or low dosages of DOX (0.2 and 1.5 µM, as DOX_L_ and DOX_H_, respectively) and DDP (5 and 30 µM, as DDP_L_ and DDP_H_, respectively). The cells were seeded and cultured overnight, and then treated by PD, DOX_L_, DOX_H_, DDP_L_ and DDP_H_ individually or combinatorially (PD + DOX_L_ and PD + DDP_L_) for 48 h. The cells were stained with Annexin V-FITC and propidium iodide, then analyzed for apoptotic cell ratios using flow cytometry. Each sample was analyzed in triplicate, with similar results observed and representative data presented. The results are shown as the mean ± S.D. * *p* < 0.05, ** *p* < 0.01, *** *p* < 0.001, ns: not significant.

**Figure 4 cells-12-00222-f004:**
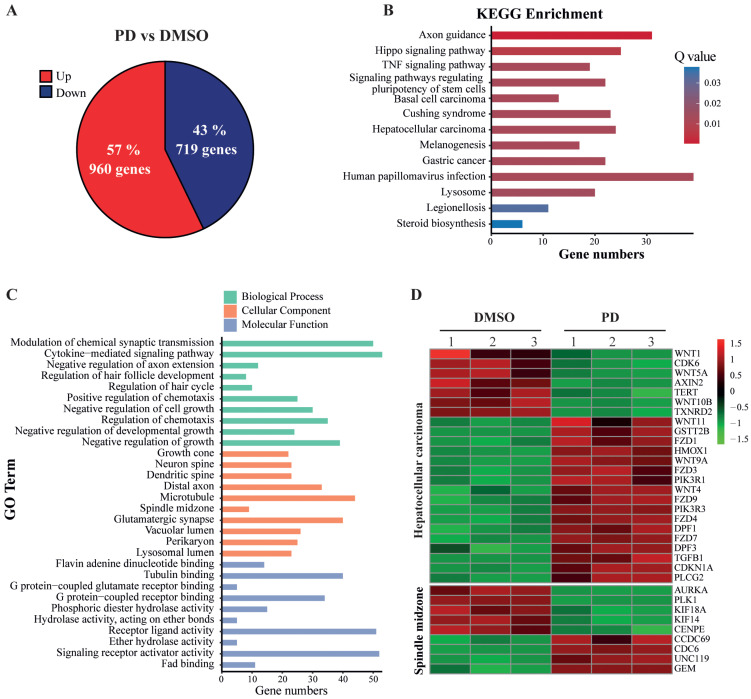
Analysis of differentially expressed genes based on RNA-seq data of HepG2 cells treated with PD. (**A**) Up- and down-regulated genes in PD-treated HepG2 cells. Collectively, 1679 genes were significantly and differentially expressed, with 960 genes upregulated (red) and 719 genes downregulated (blue) in PD-treated HepG2 cells compared to the DMSO-treated cells. (**B**) The altered signaling pathways in the KEGG enrichment analysis. (**C**) Gene Ontology (GO) enrichment of differentially expressed genes. GO contains three independent GO categories. The three indicated processes include the biological process (BP), molecular function (MF) and cellular component (CC). (**D**) The heatmap of the DEGs involved in hepatocellular carcinoma (HCC) development and spindle midzone formation.

**Figure 5 cells-12-00222-f005:**
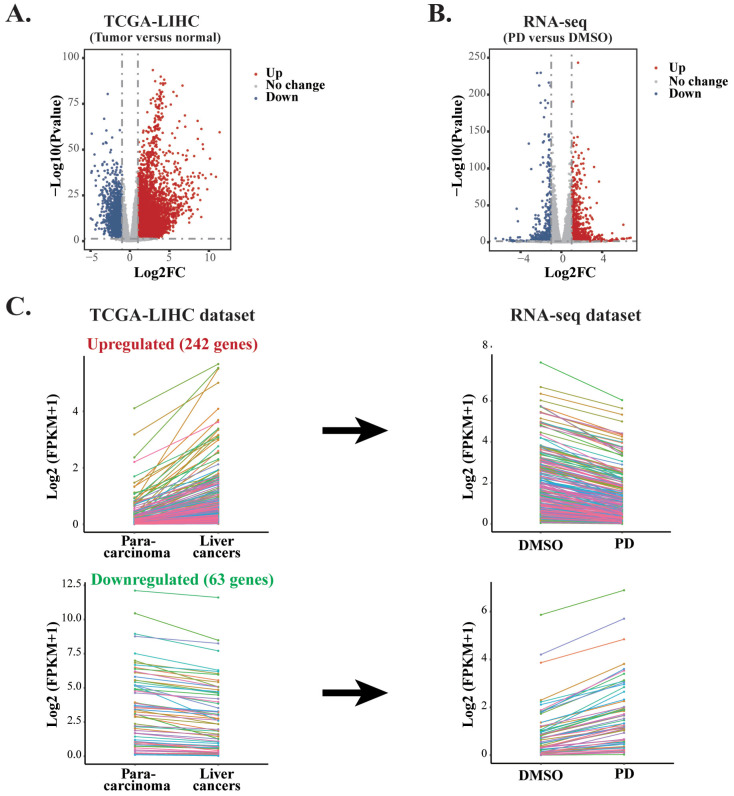
Comparative analyses of two RNA-seq datasets from PD-treated HepG2 cells and TCGA-LIHC. (**A**) The volcano plot of the DEGs in the TCGA-LIHC dataset containing 6835 upregulated genes and 1742 downregulated genes between HCC tissues and normal samples. (**B**) The volcano plot of the DEGs in the RNA-seq dataset from PD-treated HepG2 cells containing 960 upregulated genes and 719 downregulated genes. (**C**) Line charts of the genes with inverse expression between the RNA-seq dataset of PD-treated HepG2 cells and TCGA-LIHC dataset. In the TCGA-LIHC dataset, 242 upregulated and 63 downregulated genes were inversely expressed in the dataset of PD-treated HepG2 cells. The colors represent the same genes in the two datasets. (**D**) GO and KEGG enrichment analyses of the inversely expressed DEGs between the dataset from PD-treated HepG2 cells and the TCGA-LIHC dataset.

**Figure 6 cells-12-00222-f006:**
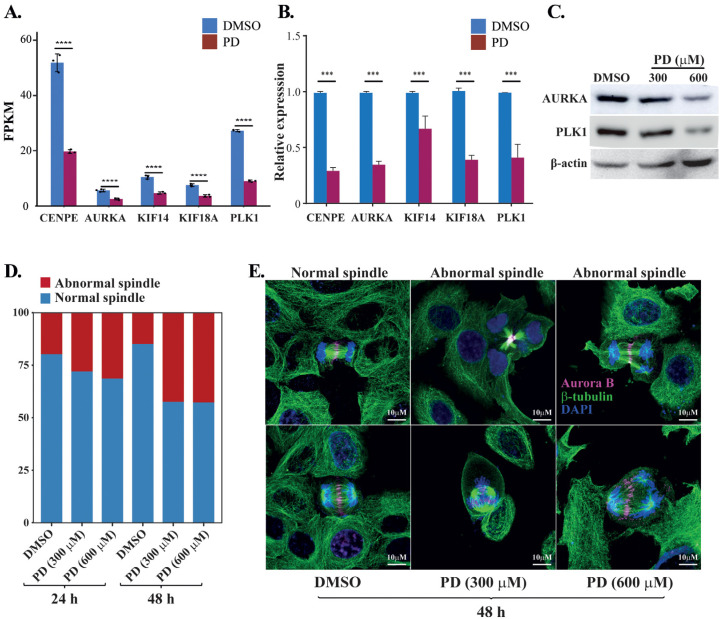
Analyses of PD-repressed genes regulating spindle midzone formation. (**A**) FPKM changes of the five genes CENPE, AURKA, KIF14, KIF18A and PLK1 regulating the spindle midzone in the RNA-seq dataset of PD-treated HepG2 cells versus DMSO-treated cells. FPKM (fragments per kilo base of transcript per million mapped fragments) is a gene expression unit to normalize counts for paired-end RNA-seq data in which two (left and right) reads are sequenced from the same DNA fragment. (**B**) RT-qPCR analyses of the five genes in the PD- and DMSO-treated HepG2 cells. In (**A**,**B**), *** denotes *p* < 0.001 and **** denotes *p* < 0.0001. (**C**) Western blot analyses of the AURKA, PLK1 and β-actin expression in the HepG2 cells treated by DMSO, 300 and 600 µM of PD. The three antibodies are against AURKA (cat# D3E4Q, Cell Signaling), PLK1 (cat# sc-17783, Santa Cruz), β-actin (cat# A5441, Sigma-Aldrich). (**D**,**E**) Analyses of spindle formation during the mitosis of HepG2 cells treated by DMSO, 300 and 600 µM of PD. In D, over 130 mitotic cells were evaluated for each treatment at 24 and 48 h time points, and the percentages of cells with abnormal and normal spindles were presented. In E, representative images of normal spindles in DMSO-treated HepG2 cells and the abnormal spindles in 300 and 600 µM PD-treated cells for 48 h. AURKB was stained in magenta showing the positions of spindle midzone, α-tubulin was stained in green showing the microtubules in spindles and DAPI was used to display the nuclei. The experiments have been repeated with the results of a similar tendency and representative data are shown.

**Figure 7 cells-12-00222-f007:**
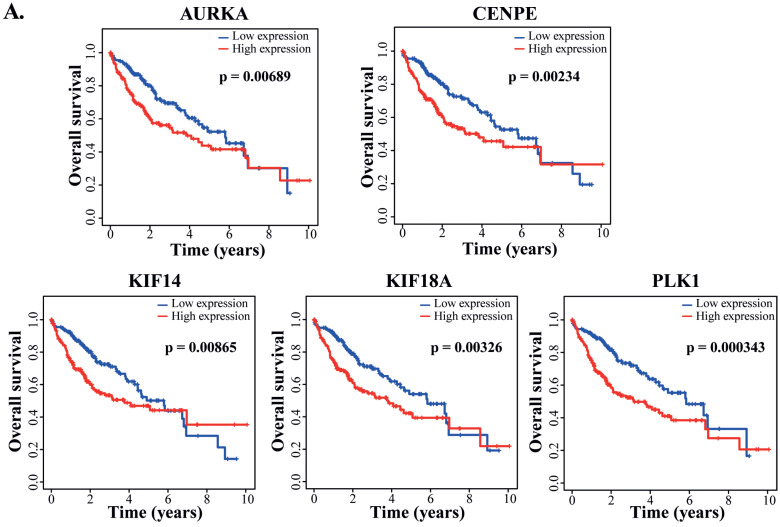
Analyses of the clinically prognostic values of PD-repressed genes regulating spindle midzone formation. (**A**) Kaplan–Meier plots of the correlation between each of the five genes regulating the spindle midzone and the overall survival rates of HCC patients in the TCGA-LIHC dataset. (**B**) The multivariate Cox’s regression analysis of the 5 DEGs regulating spindle midzone formation with the patients’ survival in the TCGA-LIHC dataset. According to the results of the Cox model, the patients were divided into low- and high-risk groups (185 and 184 patients, respectively), and Kaplan–Meier survival curves of the two groups are presented. The 5-year overall survival rates were determined as 38.4% (95% CI: 29.1%–50.8%) and 55.3% (95% CI: 44.7%–68.4%) for the high- and low-risk groups, respectively. (**C**) The receiver operating characteristic (ROC) curve to test the accuracy of the Cox model. The value of the area under the curve (AUC) was determined as 0.669, indicating a high accuracy of the Cox model analysis.

**Figure 8 cells-12-00222-f008:**
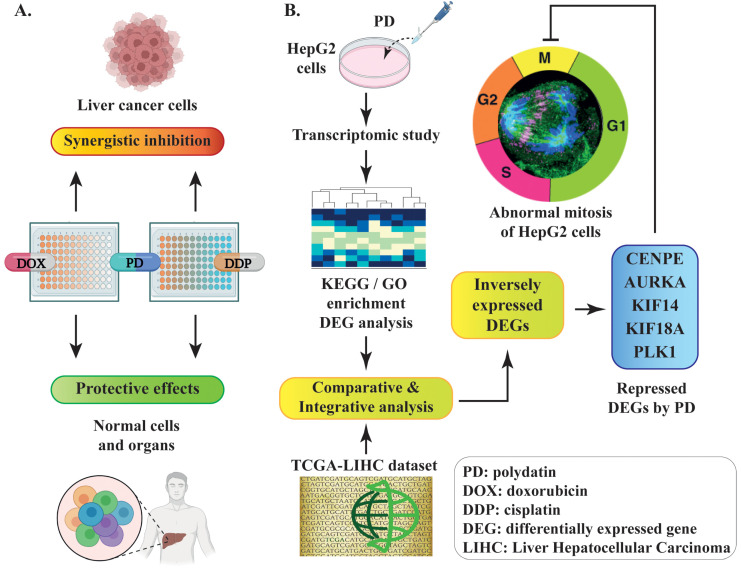
Schematic diagram of the experimental design and discovery in the current study. (**A**) PD-mediated synergistic inhibition to liver cancer cells and protective effects on normal cells when combinatorially used with DOX or DDP. (**B**) The discovery of PD-repressed DEGs related to mitosis through comparative or integrative analyses of the RNA-seq dataset from PD-treated HepG2 cells and a TCGA-LIHC dataset of liver cancer patients.

**Table 1 cells-12-00222-t001:** The IC_50_ (half maximal inhibitory concentration) values of the individual and combinatorial treatments by PD, DOX and DDP in HL-7702, HepG2 and SK-HEP1 cells for 48 h.

Drug Group		IC_50_ (μM) ± SD		
		HL-7702	HepG2	SK-HEP1
PD		872.00 ± 2.60	361.20 ± 2.67	393.00 ± 3.40
DOX		0.41 ± 0.02	1.32 ± 0.20	1.22 ± 0.07
DDP		13.30 ± 1.11	44.16 ± 1.69	25.46 ± 0.80
	PD (μM)			
DOX	50	0.33 ± 0.01	0.68 ± 0.03	0.61 ± 0.33
	100	0.32 ± 0.05	0.43 ± 0.02	0.55 ± 0.02
	200	0.38 ± 0.01	0.24 ± 0.04	0.40 ± 0.01
	250	0.44 ± 0.02	0.21 ± 0.02	0.23 ± 0.01
	300	0.45 ± 0.03	0.12 ± 0.02	0.39 ± 0.37
DDP	50	6.14 ± 0.27	15.70 ± 0.45	15.70 ± 0.96
	100	6.09 ± 0.19	11.21 ± 0.89	11.30 ± 0.30
	200	5.70 ± 0.22	6.32 ± 0.20	9.40 ± 0.56
	250	5.38 ± 0.07	3.36 ± 0.25	6.20 ± 0.33
	300	5.65 ± 0.39	2.02 ± 0.10	4.36 ± 0.28

**Table 2 cells-12-00222-t002:** The combination index (CI) values of the cotreatment of PD with DOX or DDP in HepG2, SK-HEP1 and HL-7702 cells. The CI values indicate the additive (CI = 1), antagonistic (CI > 1), or synergistic (CI < 1) effects of the treatments.

Drug Group		HepG2	SK-HEP1	HL-7702
PD (µM)				
DOX	50	0.66	0.33	0.91
	100	0.50	0.36	0.99
	200	0.45	0.34	1.10
	250	0.54	0.72	1.40
	300	0.54	0.71	1.50
DDP	50	0.41	0.69	0.84
	100	0.37	0.55	0.89
	200	0.37	0.58	1.06
	250	0.41	0.57	1.06
	300	0.51	0.63	1.25

## Data Availability

All data contained within the article.

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
