# Peer review of "Polydatin Inhibits Hepatocellular Carcinoma Cell Proliferation and Sensitizes Doxorubicin and Cisplatin through Targeting Cell Mitotic Machinery"

_cells, 2023, doi:10.3390/cells12020222_

Round 1
Reviewer 1 Report
Polydatin (PD) is a natural compound with antitumor activities against several cancer models including hepatocellular carcinoma (HCC). Doxorubicin (DOX) and cisplatin are used in HCC treatment, however toxic side effects and resistance are commonly encountered in patients. The authors studied the anticancer activities of PD against HCC cells (HepG2 and SK-HEP1) and normal-like liver lHL-7702 cells. They also evaluated by combinatorial treatments of PD with DOX and DDP in HCC cells and normal cells. They conducted RNA-seq studies and explored some genes in PD-treated HepG2 cells then used integrative analyses of the data with some differentially expressed genes (DEGs) in a TCGA dataset.
They showed that PD preferentially blocked the proliferation of HCC cells with low toxicity in normal cell. DOX or DDP combination treatments with PD synergized in HCC cells with lower effects on the normal-like liver cells. RNA-seq analysis demonstrated numerous upregulated and downregulated genes in PD-treated HepG2 cells. Integrative analyses of DEGs in PD-treated HepG2 cells and DEGs in a TCGA dataset of HCC patients indicated five PD-repressed DEGs regulating the spindle midzone formation. The expression of these genes displayed correlation with poor clinical outcomes of HCC patients. Finally, they suggest that cell mitosis may likely a primary target of PD in HCC.
The studies are interesting and well-written, although some of the experiments may not be novel. For instance, Jiao et al. have already shown that PD preferentially inhibits the growth of HCC cells versus normal-liver ones (Some of these cells are in common with the present studies) and reduced tumor growth in xenografted HepG2 cells. These studies should be referenced.
Jiao et al. Polydatin inhibits cell proliferation, invasion and migration, and induces cell apoptosis in hepatocellular carcinoma. Braz J Med Biol Res. 2018
The authors use PD concentrations ranging from 50-300 µM. This is a problem as these concentrations are not pharmacologically achievable knowing that PD has relatively low pharmaceutical effectiveness.
Does PD sensitized the antitumor effects of DOX or DDP in animal models?
The authors suggest that cell mitosis may likely a primary target of PD in HCC. The authors need to validate the protein expression of few of the five genes, in PD-treated HCC cells, that are involved in spindle midzone formation. Do the authors observe any mitotic abnormalities such as mitotic catastrophe and upregulation of cyclin B1 in PD-treated HCC cells?
Minor comments:
The authors need to describe the selected HCC cell lines.
Line 15: not clear what “reversed” means, it should be replaced by “opposite.”
Author Response
Reviewer 1:
Polydatin (PD) is a natural compound with antitumor activities against several cancer models including hepatocellular carcinoma (HCC). Doxorubicin (DOX) and cisplatin are used in HCC treatment, however toxic side effects and resistance are commonly encountered in patients. The authors studied the anticancer activities of PD against HCC cells (HepG2 and SK-HEP1) and normal-like liver lHL-7702 cells. They also evaluated by combinatorial treatments of PD with DOX and DDP in HCC cells and normal cells. They conducted RNA-seq studies and explored some genes in PD-treated HepG2 cells then used integrative analyses of the data with some differentially expressed genes (DEGs) in a TCGA dataset.
They showed that PD preferentially blocked the proliferation of HCC cells with low toxicity in normal cell. DOX or DDP combination treatments with PD synergized in HCC cells with lower effects on the normal-like liver cells. RNA-seq analysis demonstrated numerous upregulated and downregulated genes in PD-treated HepG2 cells. Integrative analyses of DEGs in PD-treated HepG2 cells and DEGs in a TCGA dataset of HCC patients indicated five PD-repressed DEGs regulating the spindle midzone formation. The expression of these genes displayed correlation with poor clinical outcomes of HCC patients. Finally, they suggest that cell mitosis may likely a primary target of PD in HCC.
The studies are interesting and well-written, although some of the experiments may not be novel. For instance, Jiao et al. have already shown that PD preferentially inhibits the growth of HCC cells versus normal-liver ones (Some of these cells are in common with the present studies) and reduced tumor growth in xenografted HepG2 cells. These studies should be referenced.
Jiao et al. Polydatin inhibits cell proliferation, invasion and migration, and induces cell apoptosis in hepatocellular carcinoma. Braz J Med Biol Res. 2018
Reply: We thank the reviewer for proposing this and apologize for not mentioning this study in our previous submission. In revised manuscript, we have discussed the work in this study and cited this paper (lines 581-584).
The authors use PD concentrations ranging from 50-300 µM. This is a problem as these concentrations are not pharmacologically achievable knowing that PD has relatively low pharmaceutical effectiveness.
Reply: We thank the reviewer for the question. Actually, many natural anticancer compounds, despite their selective or better inhibitory activity to cancer cells than that to normal cells, typically show relatively high IC50 values against cancer cells. This suggests their comparatively low effectiveness or efficacy in suppressing cancer cells, and thus impractically high dosages would be needed in their potential clinical applications. Therefore, these compounds will need to be modified to improve their anticancer efficacy, or can only be used as ancillary agents to improve the response of cancer cells to generic cancer therapeutics. We have discussed these points (lines 612-616) in the revised manuscript.
Does PD sensitized the antitumor effects of DOX or DDP in animal models?
Reply: We thank the reviewer for the question. We found that PD could synergistically increase the activity of DOX and DDP when treating the HCC cells, and it will be certainly interesting to further examine this effect in a mouse model. However, the in vivo experiment was confined by the current COVID 19 pandemic situation in our city and also the limited time of the manuscript revision. Nevertheless, the major focus of this study is the systemic investigation of PD-targeted genes and signaling pathways in liver cancer cells. We have to leave the in vivo experiments of the PD/DOX or PD/DDP cotreatment to future studies.
The authors suggest that cell mitosis may likely a primary target of PD in HCC. The authors need to validate the protein expression of few of the five genes, in PD-treated HCC cells, that are involved in spindle midzone formation. Do the authors observe any mitotic abnormalities such as mitotic catastrophe and upregulation of cyclin B1 in PD-treated HCC cells?
Reply: We thank the reviewer for the constructive suggestions. First, we used RT-qPCR to verify the repressed expression of the five genes (Figure 6B), and employed Western blot analysis to confirm the downregulation of AURKA and PLK1 expression in PD-treated HepG2 cells (Figure 6C). Other two tested antibodies somehow did not show detectable bands at the predicted positions, likely due to their poor quality. Second, as suggested by the reviewer, we carried out immunostaining experiments to evaluate whether PD could increase the aberrant mitosis of HepG2 cells. In these studies, we observed significantly increased percentage of mitotic HepG2 cells with spindle multipolarity in response to PD-treatment compared to the DMSO control (Figures 6D and 6E). Previous studies indicates that multipolar spindles, as a type of abnormal mitosis, can sensitize the cells to apoptotic stimuli according to previous literature (discussed in lines 640-647).
Minor comments:
The authors need to describe the selected HCC cell lines.
Reply: We thank the reviewer for the kind reminder. The descriptions of the employed cell lines have been added in lines 224-228 of the revised manuscript.
Line 15: not clear what “reversed” means, it should be replaced by “opposite.”
Reply: We thank the reviewer for the suggestion. The term “reversed” in the Abstract has been replaced by “opposite” (in line 19 of the revised manuscript).

Reviewer 2 Report
The authors report polydatin (PD) extracted from Chinese herbs as a natural anticancer drug, and explore its mechanisms of HCC therapy. They tested PD therapeutic efficiency in the cell lines of HepG2, SK-HEP1 and HL-7702, found PD inhibited the proliferation of HCC cells, and synergistically enhanced the therapeutic activity of DOX and DDP in HCC cells. The authors identified 1679 DEGs with over 2.0-fold changes in PD treated cells, five PD-repressed DEGs combining the result from the assay of TCGA dataset, and they concluded that PD represses cell mitosis.
Comment 1: The English writing is poor, and needs to be improved by an English native person. Such as “PD inhibited hepatocellular carcinoma (HCC)”, “we studied PD treatments”, “PD selectively blocked the proliferation of HCC cells”, and “spindle midzone formation” (midzone is not a professional term), and others……
Comment 2: At least, the expression of the five “PD-repressed DEGs” in PD treated cells should be analyzed by western blot to confirm the author’s prediction by database assay!
Comment 3: The authors presented their results by too much number data, but few photo results, especially microscopy results! For the study on the cancer therapeutic efficacy by an anticancer drug, the results about cell motility (such as Transwell), cytoskeleton polymerization (such as laser confocal microscopies, or fluorescent microscopies) cannot be absent! So the authors should add some of them.
Comment 4: The authors said that they found the mechanism and pathways by them PD functions in therapy of HCC, so they should conclude and present an illustration figure about it!
Author Response
Reviewer 2:
The authors report polydatin (PD) extracted from Chinese herbs as a natural anticancer drug, and explore its mechanisms of HCC therapy. They tested PD therapeutic efficiency in the cell lines of HepG2, SK-HEP1 and HL-7702, found PD inhibited the proliferation of HCC cells, and synergistically enhanced the therapeutic activity of DOX and DDP in HCC cells. The authors identified 1679 DEGs with over 2.0-fold changes in PD treated cells, five PD-repressed DEGs combining the result from the assay of TCGA dataset, and they concluded that PD represses cell mitosis.
Comment 1: The English writing is poor, and needs to be improved by an English native person. Such as “PD inhibited hepatocellular carcinoma (HCC)”, “we studied PD treatments”, “PD selectively blocked the proliferation of HCC cells”, and “spindle midzone formation” (midzone is not a professional term), and others……
Reply: We thank the reviewer for pointing these out for us. I deeply apologize for missing the proofreading of some parts (such as the legend of Figure 5, initially written by my students) of the manuscript prior to the initial submission. The revised manuscript has been carefully read, and any encountered grammatical and linguistic error has been corrected. Meanwhile, as suggested, the manuscript was read and edited by a native English speaker, Dr. Daniel Stovall, a tenure-tracked assistant professor at Winthrop University, Rock Hill, SC 29733, USA. Dr. Stovall was my Ph.D. student and graduated from my lab when I worked in Wake Forest School of Medicine. Dr. Stovall discovered multiple typos, which have been corrected in the revised manuscript. The “spindle midzone” is a term generated by the GO enrichment analysis of the RNA-seq dataset, and we also found that the term has been frequently used when searching it in PubMed.
Comment 2: At least, the expression of the five “PD-repressed DEGs” in PD treated cells should be analyzed by western blot to confirm the author’s prediction by database assay!
Reply: We thank the reviewer for the suggestion. We carried out RT-qPCR to verify the repressed expression of the five genes (Figure 6B), and used Western blot analysis to confirm the downregulation of AURKA and PLK1 in PD-treated HepG2 cells (Figure 6C). Other tested antibodies somehow did not show detectable bands at the predicted positions, likely due to their poor quality.
Comment 3: The authors presented their results by too much number data, but few photo results, especially microscopy results! For the study on the cancer therapeutic efficacy by an anticancer drug, the results about cell motility (such as Transwell), cytoskeleton polymerization (such as laser confocal microscopies, or fluorescent microscopies) cannot be absent! So the authors should add some of them.
Reply: We thank the reviewer for the constructive suggestion. First, we carried out the transwell assay and observed that PD-treatment could reduce the migration of HepG2 cells, shown in Figure 1C of the revised manuscript. Second, as suggested by the reviewer, we carried out immunostaining experiments to evaluate whether PD could increase the aberrant mitosis of HepG2 cells. In these studies, we observed significantly increased percentage of mitotic HepG2 cells with spindle multipolarity in response to PD-treatment compared to the DMSO control (Figures 6D and 6E). Previous studies indicates that multipolar spindles, as a type of abnormal mitosis, can sensitize the cells to apoptotic stimuli according to previous literature (discussed in lines 640-647).
Comment 4: The authors said that they found the mechanism and pathways by them PD functions in therapy of HCC, so they should conclude and present an illustration figure about it!
Reply: We thank the reviewer for the suggestion. We have added a new figure (Figure 8) to illustrate the finding of PD-related pathways in this manuscript.

Round 2
Reviewer 1 Report
The authors have adequately answered my comments.